# The Rock Garden: a preliminary assessment of how campus-based field skills training impacts student confidence in real-world field work

Thomas W. Wong Hearing[1,2], Stijn Dewaele[1], Stijn Albers[1], Julie De Weirdt[1], Marc De Batist[1]

[1]Department of Geology, Ghent University, Campus Sterre S8, Krijgslaan 281, Ghent, B-9000, Belgium
[2]School of Geography, Geology and the Environment, University of Leicester, University Road, Leicester, LE1 7RH, UK

*Correspondence to*: Thomas W. Wong Hearing (twonghearing@gmail.com); Marc De Batist (marc.debatist@ugent.be)

**Abstract.** The Rock Garden is a new on-campus field skills training resource at Ghent University, developed to increase the accessibility of and opportunities for students' geological field skills training. Developing specific field skills is integral to geoscience education and is typically concentrated into whole-day or longer field courses. These field courses have exceptional educational value as they draw together multiple strands of classroom theory and practical laboratory learning. However, field courses are expensive and time-intensive to run, and can present physical, financial, and cultural barriers to accessing geoscience education. Moreover, the relative infrequency of field courses over a degree programme means that key skills go unused for long intervals and students can lose confidence in their application of these skills. To tackle the inaccessibility of field skills training, made more pronounced in light of the coronavirus pandemic, we built the Rock Garden: an artificial geological mapping training area that emulates a real-world mapping exercise in Belgium. We have integrated the Rock Garden into our geological mapping training courses and have used it in partial mitigation of coronavirus travel restrictions. Using the Rock Garden as a refresher exercise before a real-world geological mapping exercise increased students' confidence in their field skills, and students whose education was disrupted by the coronavirus pandemic produced work of a similar quality to students from pre-pandemic cohorts. Developing a campus-based resource makes field training locally accessible, giving students more opportunities to practise their field skills and, consequently, more confidence in their abilities.

## 1 Introduction

Field work has been considered an essential component of geoscience education since the beginning of formal geoscience teaching over 150 years ago (Butler, 2008; Chiarella and Vurro, 2020; Lawrence and Dowey, 2022; Whitmeyer et al., 2009). Until recently, the Geological Society of London required up to 37 and 60 field days for accreditation of 'Geoscience' and 'Geology' undergraduate degrees, respectively (Geological Society Regulation R/FP/8; Giles et al., 2020), though this requirement was removed in a July 2023 update. In the USA, field camps are compulsory for many geology degree majors (e.g. Abeyta et al., 2021) and "general field methods" rank as the fourth most commonly required courses in geology degrees (Klyce and Ryker, 2022). Field training is typically delivered as whole-day or longer residential field courses as universities

are, unfortunately, rarely sited on the geology to be studied. These intensive field courses have exceptional educational value, drawing together myriad strands of classroom theory and practical laboratory learning in the dynamic environment of student-led discovery in tackling real-world geoscience questions (e.g. Butler, 2008; Waldron et al., 2016). Although many graduates who remain within the geosciences, including those in academia, will go on to long and distinguished careers that do not involve any field work for themselves, they will be working with data gathered in the field. As well as providing training in

field data collection methods, field courses are also useful for teaching the fundamental limitations of field data, such as uncertainty about the regional applicability of local measurements, the exact stratigraphic precision of geochemical samples, or the reasonableness of interpolating between outcrops. Understanding the practical aspects of field data collection is important for anyone who works with data collected in the field, and this includes most geoscientists.

Long, typically residential, field courses are financially costly and time-consuming to organise and run, and exert additional

pressures on teaching staff (Tucker and Horton, 2019). Consequently, field training is typically delivered in intensive but infrequent bursts throughout a degree programme. From a student perspective, the infrequency of field courses means that key skills may go unused for long periods and students can lose confidence in their field skills between courses. More fundamentally, intensive and especially long residential field courses can raise multiple barriers to accessing geoscience degree programmes (e.g. Giles et al., 2020; Tucker et al., 2022). These barriers include physical and mental accessibility concerns

(e.g. Atchison and Libarkin, 2016; Chiarella and Vurro, 2020; Greene et al., 2021; Lawrence and Dowey, 2022; Stokes et al., 2019), caring responsibilities which may make it difficult or impossible to stay away from home (e.g. Butler, 2008; Giles et al., 2020; Lawrence and Dowey, 2022; Cox et al., 2023), financial barriers from direct course and equipment costs to indirect costs due to being away from home and work (Abeyta et al., 2021), and racial, social, and cultural barriers both in student perceptions of the social environments of fieldwork and in risks to staff and student safety (e.g. Anadu et al., 2020; Demery

and Pipkin, 2021; Lawrence and Dowey, 2022; Mackay and Bishop, 2022).

Making field courses more locally accessible can mitigate some of these structural barriers to field skills training as well as giving students more opportunities for regular field skills practise. This "skills over hills" approach emphasises the training of key techniques rather than requiring travel to a specific geological site (Lawrence and Dowey, 2022, p.55). At Ghent University, we identified a need for more opportunities for field skills training before 2020, and the need to be able to provide

field work training locally came into sharper focus when travel restrictions were introduced in response to the global COVID-19 pandemic. Some educators responded to this by setting up impressive virtual field courses (e.g. Gregory et al., 2022; Peace et al., 2021; Rader et al., 2021; Senger et al., 2021; Whitmeyer and Dordevic, 2020). Virtual field courses are a valuable and viable alternative to traditional field courses, bringing specific field areas to students at home or in the classroom (Bond and Cawood, 2021). However, there are limits to the practical experience that students can gain on digital field courses (Butler,

2008), and at Ghent University we adopted an additional approach to mitigating the impact of travel restrictions based on the need to provide long-term increased access to practical field skills training. We decided to bring the geology onto campus, creating an artificial field course with real rocks – a "Rock Garden" model.

The primary aim of the Rock Garden project is to increase the accessibility of and opportunities for training key geological field skills at Ghent University. In this study, we introduce the Rock Garden and assess how incorporating an on-campus refresher of key geoscience skills into an existing field course impacts the development of students' confidence in their practical skills throughout that course.

## 2 A field course on campus

### 2.1 Previous work

We came to the notion of a field course on campus in response to local concerns, but we are far from the first to arrive at this solution to increasing field course accessibility. Several variations on the 'Rock Garden' theme have been developed for teaching and outreach activities since at least the 1960s (Table 1), particularly in Canada and the USA. Many of these have been developed in a 'boulder garden' style, with suites of isolated blocks used to showcase the geology of an area. Examples of the boulder garden variety include the Peter Russell Rock Garden at the University of Waterloo, Canada, which opened in 1982 (Peter Russell Rock Garden, 2021), and the Geology Garden, University College Cork, Ireland (Geology Garden, 2021). Boulder gardens have also been developed to preserve and showcase the regional geology of sites recognised for their important heritage, such as the Kiecle region of Poland (Górska-Zabielska, 2021) and The Rock Garden "Geologist Juan Paricio" from the Maestrazgo Geopark in Spain (Moliner and Mampel, 2019).

There are also several examples of campus-based geological teaching resources that focus on field skills training. For example, Dillon et al. (2000) describe the "Geologic Rock Garden" at the University of Western Ontario, Canada, which has a geological mapping component. Calderone et al. (2003) describe "GeoScape: an instructional rock garden" at Glendale Community College, USA, that uses a combination of coloured gravels and larger boulders to make an artificial mapping area. One of the most extensive, and perhaps longest running, examples is the Central Michigan University's "CMUland" or "Campus Geological Area". Started in the 1960s as a suite of erratic boulders, this resource has been reshaped to form a fairly complex geological mapping exercise with blocks that emulate natural outcrops (Benison, 2005; Matty, 2006).

Waldron et al. (2016) discussed recent developments at "The Geoscience Garden" at the University of Alberta, Canada, which is probably the development most similar to Ghent University's Rock Garden. The University of Alberta Geoscience Garden includes large blocks dug into the ground to create an artificial and semi-realistic geological mapping area across campus and also includes an outreach component with QR codes signposting members of the public to more information about each rock (Waldron et al., 2016).

### 2.2 Developing the Rock Garden at Ghent University

Notably, most rock gardens and particularly those in North America have been developed on universities with open-plan, spacious, campuses (Table 1; e.g. Waldron et al., 2016). Belgium is famously rather more compact. The Ghent University Geology Department is located on Campus Sterre, slightly outside of the main urban area of the city of Ghent, and has some

green space reserved for encouraging the growth of high biodiversity-supporting habitats (Biodiversiteitspad op Campus

Sterre, 2021). We worked with the Faculty of Sciences (FWE) and the Directie Gebouwen en Facilitair Beheer (DGFB, the university's estates management division) to plan artificial outcrops sympathetically with existing utilities lines, future plans for campus development, and the Campus Sterre biodiversity plan (Biodiversiteitspad op Campus Sterre, 2021). We identified the key field skills that we aimed to provide more training in, notably orienteering, identifying and measuring planar features, recording spatial data in the field, and inferring geological structures from sparse outcrops (i.e. where the whole structure

cannot be easily visualised in the available outcrop).

These skills coalesce in geological mapping exercises which have long been used as holistic geological training activities (Butler, 2008). The initial aim of the Rock Garden project, therefore, was to create a geological mapping area with different mappable units ('formations') and identifiable planar surfaces, particularly bedding, that could be used to practise taking structural measurements.

Following a suite of criteria developed with the FWE and DGFB We identified 10 areas of Campus Sterre available for development into Rock Garden outcrop sites. We produced an idealised geological plan for the Rock Garden mapping area which was constrained by the size and distribution of the sites available. Working with a local quarry, SAGREX Quenast, and a local building stone company, Monument NV, we sourced a variety of large Belgian rocks. It was important for us that the rocks were locally sourced, in keeping with the feel of the Rock Garden as an 'inland Belgian' mapping exercise.

We dug the blocks into the ground at the 10 outcrop sites to make the sites emulate bedrock cropping out of the ground rather than isolated erratic boulders (Figure 1). We agree with Matty (2006) and Waldron et al. (2016) that it is important for the installation to take place under the guidance of the designing geologists to ensure that blocks are oriented as closely as possible to the planned strike and dip measurements (Figure 1). Before teaching began, we mapped the Rock Garden outcrops and where necessary adjusted the blocks to make sure that the outcrops could be interpreted as a geologically plausible structure.

The completed Rock Garden comprises six mappable lithological units spread across 10 outcrops over an area of approximately 3.75 hectares. The Rock Garden outcrops are arranged around the eastward plunging 'S8 anticline' (Figure 2; named for the S8 building through which its axis runs). The core of the anticline comprises sandstones, shales, and limestones of Cambrian to Carboniferous age. An angular unconformity separates the youngest of these Palaeozoic deposits from Eocene volcanics and fossiliferous sandstones. In the southern part of the Rock Garden, there is evidence of an intrusive igneous lithology

discordant with the rest of the strata.

In parallel with the physical installation, we developed a 'field guide' and website to support teaching and learning on the Rock Garden. We prepared the field guide in a similar manner to that for a real-world field course, including general field work resources and site-specific information. Alongside the field guide is a website (https://www.rockgarden.ugent.be) that includes sections for staff, students, and members of the public. In particular, there are sections on the Rock Garden outcrops

and lithologies that can be used as prompts for educators (restricted to Ghent University staff, contact the authors for access), and downloadable resources for mapping exercises. Each outcrop has a small wooden signpost with a QR code that links to the Rock Garden website (Figure 3). The QR codes are primarily intended to aid learning, and the landing pages from each

outcrop have directions to access the results of field tests students might wish to perform, such as applying hydrochloric acid to the test carbonate content of the rocks. This allows students to work through the intellectual exercises of field tests without depleting the limited outcrops for future generations. The QR code landing pages also link to a non-technical explanation of the Rock Garden for interested members of the public, who have free access to the campus.

## 2.3 Teaching with the Rock Garden

We began using the Rock Garden for teaching in academic year 2020/21 as a partial replacement of undergraduate field courses in mitigation of coronavirus travel restrictions. Initial teaching activities were skills-focused, with students using isolated outcrops to identify and measure bedding planes on their first field assignment (Figure 4). We have since used the Rock Garden as an integral part of our undergraduate geological mapping training. The Rock Garden field mapping exercise takes approximately half a day, with a further half- to full-day classroom exercise to produce a finished geological map, cross section, generalised vertical section, and 'geological history'. Similar to real-world geological mapping, the limited size and number of outcrops means that there is no single 'correct' solution for the Rock Garden's geology, but rather several plausible options. We have enjoyed discussing the possible solutions with students, and what additional data they might want to collect to decide between their different hypotheses.

In addition to field course teaching, the palaeontology of the Rock Garden was the topic of a bachelor thesis project that provided training and experience in conducting a full field-based palaeontological research project on campus. The results of this project will be incorporated into a technical palaeontological annex to the Rock Garden teaching resources. This project demonstrates the potential of on-campus resources for holistic teaching of the methods and processes of field-based geoscience research.

## 3 Aims and hypotheses

The overarching goal of our study is to assess the impact on students of a local accessible field skills refresher training course prior to a real-world field exercise. As we aim for a better understanding of how students' self-perceived confidence in their abilities (self-efficacy) changes over a module, we primarily interact with the affective domain of Bloom's taxonomy of education (Krathwhol et al., 1964), which includes aspects of values, attitudes, and emotions towards learning. There is considered to be a positive link between student self-efficacy (affective response) and realised attainment (cognitive response) for a given task (e.g. Boyle et al., 2007; Stokes and Boyle, 2009; McConnell and van Der Hoeven Kraft, 2011; Mogk and Goodwin, 2012; Wiggen and McDonnell, 2017), and geoscience teaching staff have been shown to recognise that student motivation is of primary importance in influencing learning outcomes (Markley et al., 2009). Moreover, geoscience field work also seems to produce a general positive affective response in students (e.g. Mogk and Goodwin, 2012; Streule and Craig, 2016; Waldron et al., 2016) summarised neatly in the title of Boyle et al. (2007): "Field work is good". Despite its acknowledged importance for students' learning experience and outcomes, however, the affective domain remains

substantially under-studied in the geosciences, partly because it is difficult to assess (McConnell and van Der Hoeven Kraft, 2011).

Here, our primary aim is to evaluate how students' self-efficacy changes over the duration of a field course that comprises an initial refresher component in a familiar setting, the Rock Garden, and a subsequent real-world field exercise, following a similar approach to that of Waldron et al. (2016). Our secondary aim is to understand the impact of student confidence on assessed performance. In pursuit of these aims, we evaluate three hypotheses which are set out below. We evaluate the first two hypotheses for each field skill individually as well as for aggregated confidence across all field skills.

**Hypothesis 1: students' confidence will increase with the additional training**

Our first hypothesis is that students will become more confident in applying practical field skills in a new real-world area after a short refresher exercise on the artificial Rock Garden site. We are confident that our second-year undergraduate students know how to apply key field skills like compass measurements from their previous field trips, but we think that some of them lack the confidence born of familiarity when applying their skills in unfamiliar settings. A short refresher in a familiar setting should help increase their confidence when faced with a real-world challenge. We test this hypothesis by comparing students' self-reported confidence in specific skills before and after working on the Rock Garden. The null hypothesis is that there will be no significant difference in students' confidence in applying their field skills after working on the Rock Garden.

**Hypothesis 2: an artificial course will provide a greater confidence boost than a real-world exercise**

Our second hypothesis is that the greatest increase in student confidence will be due to the Rock Garden exercise, before students go into a real-world field setting. We think it is likely that a short refresher of field skills on the Rock Garden will result in a greater increase in confidence than transferring those skills to the real world because the Rock Garden has been designed as a training course and is therefore simpler than a real-world site. We test this hypothesis by examining how confidence scores change between the pre-course baseline and post-Rock Garden questionnaires, and the post-Rock Garden and post-course questionnaires. The null hypothesis is that there is no significant difference in the magnitude of self-efficacy change after the Rock Garden and real-world exercises.

**Hypothesis 3: increasing confidence increases student performance**

Our third hypothesis is that students' increased confidence in their field skills will translate into better performance in the field. We test this hypothesis by comparing marks for the Geological Mapping A module for cohorts before and after developing the Rock Garden (pre-2020 and post-2020 cohorts, respectively). Note that marking of the module Geological Mapping A was conducted by J.D.W., M.D.B., and S.A. This hypothesis was developed by T.W.W.H. and was not discussed with J.D.W., M.D.B., or S.A. before marking was completed. The null hypothesis is that there is no significant difference in attainment between pre-2020 and post-2020 cohorts.

## 4 Methods

This study was conducted in accordance with the General Ethical Protocol for Scientific Research at the Faculty of Psychology and Educational Sciences (FPPW) of Ghent University and was reviewed by the FPPW Ethics Committee. For this study of the impact of the Rock Garden, we compiled and assessed two datasets. To produce the first dataset, we conducted a series of questionnaires in which students self-assessed their confidence in conducting geological field work during their Geological Mapping A field course in academic year 2021/22. We distributed identical anonymous self-assessment questionnaires (Table

2) to all students on the 2021/22 Geological Mapping A course (n = 14). Undergraduate education at Ghent University is conducted in Dutch so the questionnaires were also conducted in Dutch; the English translations are provided here. Identical questionnaires were distributed on three occasions throughout the course:

1. a pre-course questionnaire at the start of the course ("Pre");
2. a mid-course questionnaire, after students had completed their Rock Garden work ("Mid"); and

200 3. a post-course questionnaire, after students had completed their 'real-world' field work ("Post").

The questionnaires comprised a list of 10 geoscience skills that we aim to develop throughout the 'Geological Mapping A' module. Students were asked to self-assess their confidence in applying each of these skills on a five-point Likert scale (1: not at all confident; 5: very confident). Questionnaires were distributed and collected by S.A. and the responses were independently digitised and analysed by T.W.W.H.. The responses we received are summarised in Table 3. In addition to this quantitative

dataset, we solicited qualitative feedback on the Rock Garden from students in both the 2020/21 and 2021/22 cohorts after completion of their courses. These qualitative comments were reviewed in light of the quantitative data analyses.

The second dataset comprised anonymised marks from students on the Geological Mapping A (second year) course in cohorts before the Rock Garden was built (academic years 2015/16 to 2018/19), and after the Rock Garden was built (academic years 2020/21 to 2021/22). This dataset includes the marks of 60 pre-Rock Garden students and 28 post-Rock Garden students (Table

5). Marks for the 2019/20 cohort have not been included here because COVID-19 restrictions in Belgium meant that there was no practical field component for these students; Geological Mapping A is taught in Semester 2 and had to be changed to a fully classroom-based course at short notice due to societal lockdowns. The post-Rock Garden results are split into "Rock Garden" and "Field" components. Ongoing COVID-19 restrictions meant that a real-world "Field" component was not possible for the 2020/21 students (n = 14), who instead were taught with a combination of Rock Garden exercises and a classroom-based

virtual field course. Therefore, the post-Rock Garden "Field" data are only available for the 2021/22 cohort (n = 14). Because of the small cohort sizes, to protect student anonymity, we have not included the raw data of this compilation here. Instead, this dataset is summarised in Table 5.

All quantitative analyses were conducted using the statistical software R (R Core Team, 2021). Welch two-sample t-tests were performed using the base R *stats* package; sample sizes, t-statistics, and *p*-values are reported for each test. The small cohort

sizes in Ghent University Geology, whilst being beneficial for teaching and learning, constrain the statistical power of this study and means that the statistical analyses should be interpreted cautiously.

## 5 Results

In this study, we set out to test three hypotheses:

1. students' confidence in their practical field skills will increase following a refresher exercise on the Rock Garden (null hypothesis: there is no significant difference in student confidence following the Rock Garden exercise);

2. students will gain a greater confidence boost from the Rock Garden exercise than from a real world field course (null hypothesis: there is no significant difference between confidence changes after Rock Garden and real-world exercises); and

3. students who had a refresher exercise on the Rock Garden (post-2020 cohorts) will perform better over the whole Geological Mapping A course than students who took the module before the Rock Garden was developed (pre-2020 cohorts) (null hypothesis: there is no significant difference in attainment between pre-2020 and post-2020 cohorts).

The first two hypotheses can be addressed with the questionnaire dataset; the third hypothesis can be tested using the students marks dataset.

The results of the questionnaires are summarised in Table 3, Figure 5, and Figure 6, and the results of Welch two-sample $t$-tests comparing student confidence before, during, and after their course are presented in Table 5. We received questionnaire responses from the whole 2021/22 Geological Mapping A cohort for the first two ("Pre" and "Mid") questionnaires (n = 14), and from all but one of the students for the final ("Post") questionnaire (n = 13). In the first questionnaire, one student did not answer question *(d: measuring strike & dip)*, and one student did not answer question *(g: completing a map)*.

Mean student confidence in all skills questioned increased over the duration of the course (Table 3; Figure 5) and, with the exception of *(a)* locating oneself on a map and *(j)* conducting independent field work, mean student confidence increased after both the Rock Garden and real world components of the field course (Figure 6). Comparing between the pre- and post-course questionnaires, i.e. over the whole course, student confidence improved for all skills (at the 95 % significance level) except *(a)* locating oneself on a map and *(h)* constructing cross-sections (Table 5). There was no significant increase in student confidence in these two skills across any steps in the Geological Mapping A module.

Considering only the Rock Garden part of the course, students' confidence increased significantly at the 95 % level for six of the 10 questions *(d–g, i–j)* and for an aggregate of all confidence scores following the Rock Garden exercise (Table 5). We can therefore reject the null hypothesis at the 95 % significance level for hypothesis 1 and suggest that students' confidence in their field skills increases from a short refresher exercise on the on-campus Rock Garden. However, this is not a uniform or uniformly significant increase and it is instructive to consider which skills did or did not receive a significant confidence increase following the Rock Garden exercise.

Aggregating scores across all questions, student confidence increased significantly ($p \ll 0.001$) after both Rock Garden and real-world exercises (Table 5). This is not particularly surprising as we would hope and expect students gain confidence from practising their skills; however, it is instructive to interrogate the differences in response between specific skills.

Student confidence in *(a)* locating oneself on a map, *(b)* lithology identification, and *(c)* identifying planar surfaces increased more following the real-world exercise than from the Rock Garden exercise (Table 3; Figure 6). However, only *(b)* lithology identification and *(c)* identifying planar surfaces show no statistically significant increase in confidence after the Rock Garden exercise (*b. lithology identification*: $t(22.29) = -0.71$, $p = 0.49$; *c. planar surfaces*: $t(26.00) = 0.92$, $p = 0.36$) but do show a significant confidence increase after the real-world exercise (*b. lithology identification*: $t(22.73) = -2.16$, $p = 0.04$; *c. planar surfaces*: $t(17.41) = -3.45$, $p < 0.01$; Table 5). In all other skills, there was a greater increase in confidence (a stronger self-efficacy response) after the Rock Garden exercise than after the real-world exercise. The highest confidence in all areas was achieved after the full course, including the real-world exercise (Figure 6). This includes *(d)* making and *(f)* plotting strike and dip measurements, where there was a significant increase in confidence following the Rock Garden exercise (*d. making measurements*: $t(19.64) = -2.63$, $p = 0.02$; *f. plotting measurements*: $t(21.56) = -3.58$, $p < 0.01$) but an insignificant increase in confidence following the real-world exercise (*d. making measurements*: $t(23.60) = -0.79$, $p = 0.44$; *f. plotting measurements*: $t(24.67) = -1.14$, $p = 0.27$; Table 5). We can therefore also reject the null hypothesis at the 95 % significance level for hypothesis 2 and suggest that students gain more confidence in their field skills following the on-campus Rock Garden exercise than the real-world exercise, although there is variation in the confidence boost to different skills at each interval.

The student marks dataset is summarised in Table 4 and Figure 7, and the results of Welch two-sample *t*-tests comparing pre- and post-Rock Garden marks are presented in Table 6. Marks for the field components of the course (Table 4; Figure 7) are slightly higher for post-Rock Garden cohorts ("Field" mean = 14.71, standard deviation [s.d.] = 1.82; "Rock Garden" mean = 15.14, s.d. = 1.92; "Field + Rock Garden" mean = 15.00, s.d. = 3.64) than for the pre-Rock Garden cohorts ("Field" mean = 14.48, s.d. = 1.70). Following Welch two-sample *t*-tests, there are no significant differences at the 95 % confidence interval between the pre-Rock Garden and post-Rock Garden marks for any of the components of the course (Table 6). We must therefore accept the null hypothesis for hypothesis 3: a refresher exercise on the Rock Garden that does increase student confidence in their field skills does not significantly improve student marks across the whole course.

## 6 Discussion

We developed the Rock Garden as a field course on campus with the overall aims of increasing the accessibility of geoscience field skills training for current and future students, and consequently increasing our students' confidence in applying their skills in the real world. Our preliminary results presented here suggest that incorporating the Rock Garden into geological field skills teaching through the Geological Mapping A module has delivered a positive affective response in our students, supporting previous research findings on the use of on-campus field skills training resources (e.g. Benison, 2005; Waldron et al., 2016). The small sample size of our study means that statistical results should be interpreted cautiously, but our initial results are promising and suggest that the use of artificial geological training resources is a potentially important teaching innovation and that there is wide scope for further research in this area.

Our experience using the Rock Garden for teaching is that students take well to both the letter and the spirit of the artificial exercises. Indeed, it was particularly gratifying to see that students were still enthusiastic for practical field work following pandemic travel restrictions, a sentiment encapsulated by one student who wrote that:

*"Personally, I thought it was a very cool experience. Perhaps it also had to do with the fact that we were finally allowed to do an on-campus activity with other fellow students, after a year of little fieldwork and social contacts"* — [translated from

290 Dutch].

Addressing our first and second hypotheses, (1) students' confidence increased significantly in six of the 10 skills and across an aggregate of all skills (Table 5), and (2) students' confidence increased more from the Rock Garden exercise than from the real-world exercise in all but three skills. We could therefore reject the null models for hypotheses 1 and 2. We also think that students carry their increased confidence obtained through the Rock Garden exercise through to the real-world field work

because there is a second increase, never a decline, in skills confidence following the real-world exercise (Figure 6). This is also reflected in qualitative feedback including from a student who wrote that they were:

*"convinced that before the real mapping [exercise] in the Hoyoux Valley starts it is useful to have already done a 'simpler' exercise with fewer outcrops"* — [translated from Dutch]

Overall, our findings from the first two hypotheses support previous research on students' affective response to an artificial

field training resource. Benison (2005) and Waldron et al. (2016) examined the student experience of incorporating into geological field teaching the Central Michigan University Campus Geological Area, or 'CMULand', and the University of Alberta Geoscience Garden, respectively. Benison's (2005) results showed that students found the CMULand exercise improved their understanding of general concepts covered in the 'Earth History' module, as well as improving specific field skills, like rock and fossil identification, and general skills, like teamwork. Waldron et al. (2016) asked student cohorts both

before and after the Geoscience Garden was incorporated into teaching "how useful was [module] EAS 233 in preparing you for the following aspects of [the subsequent module] EAS 234?", with a list of five specific field skills along with an assessment of overall preparedness (Ibid, fig. 8). Waldron et al. (2016) found that students felt that module EAS 233 was better preparation for the subsequent field module EAS 234 when it included the Geoscience Garden exercise. As well as an overall positive response, Waldron et al. (2016) reported positive skews in specific field skills following use of the Geoscience Garden.

Whilst our results echo the broadly positive affective response in student learning from an on-campus field resource found by previous researchers (Benison, 2005; Waldron et al., 2016), the aggregated results mask some complexity and variation between different field skills. Four field skills (*a. location on a map, b. lithology identification, c. identifying planar surfaces,* and *h. constructing cross-sections*) showed a qualitative but insignificant increase in confidence following the Rock Garden exercise (Table 5) and three of those (*a. location on a map, b. lithology identification,* and *c. identifying planar surfaces*)

showed a greater increase in confidence from the real-world exercise (Figure 6). It is instructive to consider these four skills in more detail here.

Two of these skill areas showed no significant increase in confidence across the whole course: *(a)* locating oneself on a map, and *(h)* constructing cross-sections (Table 5). Student confidence in *(a)* locating oneself on a map started from a high baseline

(mean = 3.9, s.d. = 1.4; Table 3) which may explain the statistically insignificant increase in confidence that reflects a shortening lower tail of the distribution in the "Mid" questionnaire. Similarly, student confidence in *(h)* constructing cross-sections started from a higher-than-average baseline confidence (mean = 3.2, s.d. = 0.7; Table 3) and so had a smaller potential for increase than other skills. Nevertheless, there was only a modest increase in confidence at constructing cross-sections over the whole course ("Post" mean = 3.7, s.d. = 0.6; Table 3), perhaps indicating that we have more work to do in developing students' confidence and understanding of constructing geological cross-sections through this or other courses. Interestingly, Waldron et al. (2016) also found that students who worked on their Geoscience Garden thought "they could have been more prepared in the skills of making a geologic *[sic]* cross-section" (Ibid, 227). This perhaps points to a broader challenge and area for further research in geoscience teaching of how to better develop practical skills in constructing geological cross-sections.

The two other skill areas that show no significant increase in confidence after the Rock Garden exercise, *(b)* identifying lithologies in the field and *(c)* identifying planar surfaces, do show statistically significant increases following the real-world exercise (Table 5). We think that this is due to the contrast between the Rock Garden's limited range of lithologies, which can in any case be readily differentiated, and the greater variety and complexity of lithologies encountered in the real-world setting, meaning that students get more practise, and therefore confidence, in lithology identification in the real-world exercise. Similarly, the planar surfaces of the Rock Garden are readily identifiable and may not provide as substantial a training experience as the planar features found in the real-world setting. Consequently, we think that real-world and artificial field exercises offer different opportunities for skills training and should be considered as complementary, rather than competing, exercises.

Our results regarding student self-efficacy qualitatively support the findings of previous work with campus-based geoscience field training resources in that students find such resources helpful training in general, and that it is beneficial for subsequent field work (Benison, 2005; Waldron et al., 2016). Our work provides some additional nuance to this topic, pointing to field-skill-specific variation in the timing of self-efficacy gains throughout the learning experience; some skills may see a stronger response from the on-campus exercise while other skills benefit more from the real-world exercise. We are wary of taking our discussion on this point too far, given the small numbers in our study, but would like to see future work more fully test this variation in how, when, and where specific field skills may be best developed.

Addressing our third hypothesis: there was no significant difference in students' performance in Geological Mapping A between pre- (2015-19) and post- (2020-22) Rock Garden cohorts (Table 6; Figure 7). So far, we do not have evidence that increasing students' confidence in their field skills has delivered an increase in field geology performance, and we therefore accepted the null model for hypothesis 3. Whilst there was a very slight increase in marks for the field component of Geological Mapping A, this was statistically insignificant ($p = 0.66$ for "Field – Field"; $p = 0.12$ for "Field – Field + Rock Garden") and there was similar magnitude decrease in performance in the report part of the course over the same time. It remains to be seen whether future cohorts whose field education is less disrupted will see an improvement in course marks from the development of the Rock Garden.

We would like to note, however, our results are consistent with those of Lundmark et al. (2020), who showed that whilst use of a digital field work tool delivered a substantial increase in self-efficacy, there was no concomitant increase in assessed performance in the field. This result is contra previous studies and broader theory in which impacts in the affective and cognitive domains are closely linked (e.g. Mogk and Goodwin, 2012). There are a number of possible confounding factors, at least in our study, that make this a difficult problem to address, including the subjectivity of marking performance in the field and changes in field conditions from year to year. We make no conclusions on this point, but do suggest that the link between affective and cognitive responses to field teaching innovations deserves further investigation.

The utility of the Rock Garden notwithstanding, we do not consider it to be a substitute for our established field courses. Rather, the Rock Garden is complementary and can be employed for introductory and refresher field skills training, increasing students' confidence applying their skills in subsequent real-world field exercises. Students can then make the most of their time on real-world exercises, applying their skills with confidence to ask and answer more searching questions of the geological settings they are working in.

## 7 Future developments of the Rock Garden

The Rock Garden and associated resources were completed in early 2021. Although presently limited by spatial constraints, with the geological structure now established we are in a good position to add new outcrops if and when additional campus space becomes available. This development at Ghent University shows that the 'field course on campus' model can be adapted successfully to institutions with tighter spatial constraints, and can be implemented with minimal intrusion on other planned developments. The initial aim was to provide a local, accessible, venue for teaching and practising key geological field skills. However, there is broader potential for training in field-based project methodologies, from experimental design, to field work and data collection, to laboratory analysis and interpretation. We do not consider the Rock Garden an ideal setting for such a project, but it does provide a local and accessible setting for field-based studies in petrology and/or palaeontology. With further development, the Rock Garden could be a suitable venue for routine field-based project training within our degree programmes that is robust to accessibility concerns.

The local and more controlled environment of the on-campus Rock Garden is a way to reduce accessibility barriers to teaching field geology. The area of Campus Sterre is more physically and financially accessible than many real-world field localities and field hazards can be carefully managed so that expensive field safety gear (e.g. good boots) is not required for an introduction to geological field work. This makes it possible to introduce prospective and new students to field work when the various barriers may previously have been very difficult to navigate. The Rock Garden creates the potential to introduce pre-university students to some of the practical aspects of field geology, helping to shed some light on what practical geology field work can be like. As at many universities, we run on-campus experience days for local school pupils, and with the Rock Garden we can reduce some of the barriers to the first steps of gaining practical experience of field geology even before students enrol for university degrees.

Campus Sterre is open to members of the public on foot or bicycle and is used by local people for running and walking, and the Rock Garden outcrops now make prominent features on the southern part of campus. The QR codes associated with each outcrop (Figure 3) are primarily intended as teaching aids, but the landing pages from each QR code link to the main public pages of the Rock Garden website where interested members of the public can learn more about the project, the importance of field work in geoscience, and the geology of the Rock Garden. We are in the early stages of planning a "geodiversity path" that can build on the current Rock Garden and showcase aspects of Belgian geology. We hope that this geodiversity path will complement the "biodiversity path" that already exists on Campus Sterre and highlights the diversity of floral and fungal habitats on campus (Biodiversiteitspad op Campus Sterre, 2021).

## 8 Conclusions

Field skills training is an important part of a geological education, and accessibility concerns should not present insurmountable barriers to gaining that education (e.g. Abeyta et al., 2021; Giles et al., 2020; Greene et al., 2021; Lawrence and Dowey, 2022; Tucker et al., 2022). Although many geoscience graduates will have careers that never require practical application of the field skills typically developed in an undergraduate degree, anyone working in the geosciences generally and geology in particular will interact with data acquired from field work. It is important for people working with field data to have an understanding of the practicalities involved in field data collection. A campus-based resource like the Rock Garden provides one method for mitigating accessibility issues and for improving the student experience by increasing field skills training and development opportunities.

The Rock Garden that we have constructed works for our situation at Campus Sterre at Ghent University. We had the opportunity to develop several small areas of campus, and we have been able to show that an interesting and quite complex geological problem can be constructed with a relatively small number of relatively small sites. This type of training is useful for students who may need to work in or with data from outcrop-sparse regions such as inland Belgium. The exact model we have developed on Campus Sterre will not work for all campuses at all universities, but we have shown that such activities can be developed within a modest and heavily used space. Students positively engage with both the spirit and the letter of artificial geological training activities, and these activities help students to develop and maintain confidence in applying their practical skills in real-world field exercises. Artificial field courses are not a panacea, however, and there are some skills that are better trained in a real-world field setting. An artificial field course like the Rock Garden is complementary to existing real-world field courses and can make geoscience field skills education more robust to global shocks like the coronavirus pandemic by facilitating field teaching locally.

## Author contributions

T.W.W.H., M.D.B., S.D., J.D.W., and S.A. conceived and developed the initial idea. T.W.W.H. and S.D. organised and supervised installation of the Rock Garden. M.D.B., S.A., and J.D.W. led teaching work with the Rock Garden. T.W.W.H. led the writing up, with contributions and revisions from all authors.

## Competing interests

The authors declare that they have no competing interests.

## Ethics statement

This study was conducted in accordance with the General Ethical Protocol for Scientific Research at the Faculty of Psychology and Education Sciences (FPPW) of Ghent University and received an ethics waiver from the FPPW Ethics Committee.

## Acknowledgements

We are grateful to the Ghent University Directie Gebouwen en Facilitair Beheer (DGFB) for practical assistance preparing the ground and installing the Rock Garden. We are grateful to SAGREX Quenast for supplying blocks of the Quenast intrusion and to Monument NV for supplying the other blocks. Development of the Rock Garden was funded by the Ghent University Faculty of Sciences Active Learning Fund 2020-5. T.W.W.H. is supported by BOF Postdoctoral Fellowship 01P12419.

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

**Tables**

**Table 1** Comparative table of published 'Rock Gardens' from around the world.

| Name | Institution | Country | Date begun | Primary purpose | Boulder garden | Mapping component | References |
|---|---|---|---|---|---|---|---|
| CMUland (Campus Geological Area) | Central Michigan University | USA | 1960s | Teaching | X | X | Benison (2005); Matty (2006) |
| Peter Russell Rock Garden | University of Waterloo | Canada | 1982 | Outreach, Teaching | X | | Peter Russell Rock Garden (2021) |
| St Mary's Cement Rock Garden | University of Western Ontario | Canada | 1992 | Teaching | X | X | Dillon et al. (2000) |
| GeoScape: an instructional rock garden | Glendale Community College | USA | 2001 | Teaching | X | X | Calderone et al. (2003) |
| California Rock Garden | University of California, Davis | USA | 2008 | Outreach, Teaching | X | | The California Rock Garden (2021) |
| The Geoscience Garden | University of Alberta | Canada | 2008 | Teaching | X | X | Waldron et al. (2016) |
| RockWalk Park Trail | Ontario Trails | Canada | [a]<2009 | Outreach | X | | RockWalk Park Trail (2021) |
| Rock Around the University | University of Glasgow | UK | 2011 | Teaching | X | X | Curry et al. (2020); Rock Around the University (2021); Dempster (2012) |
| Fred Webb Jr. Outdoor Lab and Rock Garden | Appalachian State University | USA | [a]<2013 | Teaching | X | | Fred Webb Jr. Outdoor Lab and Rock Garden (2021) |
| Geological Garden | University College Cork | Ireland | 2013 | Outreach | X | | Geology Garden (2021) |
| The Rock Garden "Geologist Juan Paricio" | Maestrazgo Geopark | Spain | 2013 | Outreach | X | | Moliner and Mampel (2019) |
| Monash Earth Sciences Garden | Melbourne University | Australia | 2015 | Teaching | X | | Monash Earth Sciences Garden (2021) |
| William & Mary Geology Rock Garden | College of William & Mary | USA | 2017 | Teaching | X | | How Our Garden Grows: The William & Mary Geology Rock Garden (2021) |
| Geoscience Garden | University of Canterbury | New Zealand | 2018 | Teaching | X | X | Pedley (2018) |
| Rock Garden | Edith J. Carrier Arboretum | USA | 2018 | Outreach | X | | Elmi et al. (2020) |
| Geoscience Garden | State University New York | USA | 2019 | Teaching | X | X | SUNY Potsdam Installs New Geoscience Garden (2021) |
| The Rock Garden | Jan Kochanowski University | Poland | 2019 | Outreach, Teaching | X | | Górska-Zabielska (2021) |
| The Rock Garden | Ghent University | Belgium | 2021 | Teaching | X | X | This study |

[a] The oldest confirmed year of use.

**Table 2** The questionnaire that we used to evaluate students self-assessment of their confidence before Rock Garden training, after Rock Garden training, and after their real world field work. Note: this is an English translation of the original Dutch questionnaire.

| How confident are you at: | not at all confident 1 | 2 | 3 | 4 | very confident 5 |
|---|---|---|---|---|---|
| (a) locating yourself on a map | | | | | |
| (b) identifying and describing rocks in the field | | | | | |
| (c) identifying planar surfaces (e.g. bedding planes) in the field | | | | | |
| (d) making strike/dip measurements | | | | | |
| (e) plotting outcrops on a map | | | | | |
| (f) plotting strike/dip measurements on a map | | | | | |
| (g) making a complete geological map | | | | | |
| (h) making a cross section from a geological map | | | | | |
| (i) making a generalised vertical section from a geological map | | | | | |
| (j) conducting geological field work independently | | | | | |
| Any additional comments: | | | | | |


**Table 3** Summary of results of the questionnaires (Table 3). See also Figure 6.

| Question | Questionnaire | N[a] | Median (/5) | Mean (/5) | Standard deviation |
|---|---|---|---|---|---|
| *(a)* locating yourself on a map | Before Rock Garden (Pre) | 14 | 4.0 | 3.9 | 1.4 |
| | After Rock Garden (Mid) | 14 | 4.0 | 3.9 | 1.3 |
| | After real world course (Post) | 13 | 5.0 | 4.5 | 0.8 |
| *(b)* identifying and describing rocks in the field | Before Rock Garden (Pre) | 14 | 3.0 | 2.9 | 0.6 |
| | After Rock Garden (Mid) | 14 | 3.0 | 3.1 | 0.9 |
| | After real world course (Post) | 13 | 4.0 | 3.8 | 0.6 |
| *(c)* identifying planar surfaces in the field | Before Rock Garden (Pre) | 14 | 3.0 | 2.5 | 1.0 |
| | After Rock Garden (Mid) | 14 | 3.0 | 2.9 | 1.0 |
| | After real world course (Post) | 13 | 4.0 | 3.9 | 0.4 |
| *(d)* making strike/dip measurements | Before Rock Garden (Pre) | 13 | 3.0 | 3.1 | 1.3 |
| | After Rock Garden (Mid) | 14 | 4.0 | 4.1 | 0.8 |
| | After real world course (Post) | 13 | 4.0 | 4.3 | 0.6 |
| *(e)* plotting outcrops on a map | Before Rock Garden (Pre) | 14 | 3.0 | 2.5 | 1.0 |
| | After Rock Garden (Mid) | 14 | 4.0 | 3.5 | 0.8 |
| | After real world course (Post) | 13 | 4.0 | 4.1 | 0.6 |
| *(f)* plotting strike/dip measurements on a map | Before Rock Garden (Pre) | 14 | 3.0 | 2.6 | 1.1 |
| | After Rock Garden (Mid) | 14 | 4.0 | 3.9 | 0.7 |
| | After real world course (Post) | 13 | 4.0 | 4.2 | 0.7 |
| *(g)* making a complete geological map | Before Rock Garden (Pre) | 13 | 2.0 | 2.2 | 0.9 |
| | After Rock Garden (Mid) | 14 | 3.5 | 3.4 | 0.9 |
| | After real world course (Post) | 13 | 4.0 | 3.8 | 0.7 |
| *(h)* making a cross section from a geological map | Before Rock Garden (Pre) | 14 | 3.0 | 3.2 | 0.7 |
| | After Rock Garden (Mid) | 14 | 3.5 | 3.6 | 1.2 |
| | After real world course (Post) | 13 | 4.0 | 3.7 | 0.6 |
| *(i)* making a generalised vertical section from a geological map | Before Rock Garden (Pre) | 14 | 3.0 | 2.9 | 0.7 |
| | After Rock Garden (Mid) | 14 | 4.0 | 3.8 | 0.7 |
| | After real world course (Post) | 13 | 4.0 | 4.2 | 0.8 |
| *(j)* conducting geological field work independently | Before Rock Garden (Pre) | 14 | 2.5 | 2.5 | 1.1 |
| | After Rock Garden (Mid) | 14 | 3.0 | 3.3 | 0.8 |
| | After real world course (Post) | 13 | 4.0 | 3.3 | 1.0 |
| all questions | Before Rock Garden (Pre) | 138 | 3.0 | 2.8 | 1.1 |
| | After Rock Garden (Mid) | 140 | 4.0 | 3.5 | 1.0 |
| | After real world course (Post) | 130 | 4.0 | 4.0 | 0.7 |

[a] One student did not complete the final questionnaire; in the first questionnaire, one student did not answer question (d) and another student did not answer question (g).

**Table 4** Summary statistics of the student marks dataset.

| Year[a] | 2015-19 | 2020-22 |
|---|---|---|
| *Field* | | |
| N | 60 | 14 |
| Mean | 14.48 | 14.71 |
| Standard deviation | 1.70 | 1.82 |
| *Rock Garden* | | |
| N | 0 | 28 |
| Mean | NA | 15.14 |
| Standard deviation | NA | 1.92 |
| *Field + Rock Garden* | | |
| N[b] | 60 | 42 |
| Mean | 14.48 | 15.00 |
| Standard deviation | 1.70 | 3.64 |
| *Report* | | |
| N | 60 | 28 |
| Mean | 15 | 14.44 |
| Standard deviation | 1.99 | 1.35 |
| *Total* | | |
| N | 60 | 28 |
| Mean | 14.74 | 14.48 |
| Standard deviation | 0.99 | 0.68 |

[a]2015-19 = pre-Rock Garden; 2020-22 = post-Rock Garden; the 2019/20 cohort is not included (see text for details).
[b]2020-22 'Field + Rock Garden' marks comprise 28 Rock Garden marks (2020/21 and 2021/22) and 14 Field marks (2021/22 only).

**Table 5** Results of Welch two-sample t-tests on the questionnaire responses, comparing pre-, mid-, and post-course responses.

| Question | Pre-course/Mid-course | | | Mid-course/Post-course | | | Pre-course/Post-course | | |
|---|---|---|---|---|---|---|---|---|---|
| | $t$-statistic | Degrees of freedom | $p$-value | $t$-statistic | Degrees of freedom | $p$-value | $t$-statistic | Degrees of freedom | $p$-value |
| a | -0.14 | 25.92 | 0.89 | -1.28 | 21.21 | 0.21 | -1.40 | 20.53 | 0.18 |
| b | -0.71 | 22.29 | 0.49 | -2.16 | 22.73 | 0.04 | -3.66 | 24.75 | $1.2 \times 10^{-3}$ |
| c | -0.92 | 26.00 | 0.36 | -3.45 | 17.41 | $3.0 \times 10^{-3}$ | -4.68 | 17.48 | $2.0 \times 10^{-4}$ |
| d | -2.63 | 19.64 | 0.02 | -0.79 | 23.60 | 0.44 | -3.33 | 16.51 | $4.1 \times 10^{-3}$ |
| e | -2.94 | 24.04 | 0.01 | -2.14 | 24.79 | 0.04 | -4.85 | 22.08 | $7.5 \times 10^{-5}$ |
| f | -3.58 | 21.56 | $1.7 \times 10^{-3}$ | -1.14 | 24.67 | 0.27 | -4.36 | 22.24 | $2.5 \times 10^{-4}$ |
| g | -3.42 | 24.95 | $2.2 \times 10^{-3}$ | -1.56 | 23.88 | 0.13 | -5.39 | 22.48 | $1.9 \times 10^{-5}$ |
| h | -0.99 | 21.37 | 0.33 | -0.23 | 20.28 | 0.82 | -1.72 | 24.97 | 0.10 |
| i | -3.61 | 25.93 | $1.3 \times 10^{-3}$ | -1.27 | 23.94 | 0.22 | -4.56 | 23.39 | $1.3 \times 10^{-4}$ |
| j | -2.15 | 24.20 | 0.04 | -0.17 | 23.50 | 0.86 | -2.11 | 24.99 | 0.04 |
| all | -5.82 | 272.82 | $1.7 \times 10^{-8}$ | -4.09 | 258.10 | $5.8 \times 10^{-5}$ | -10.22 | 244.56 | $1.2 \times 10^{-20}$ |

**Table 6** Results of the Welch two-sample t-tests between the pre-Rock Garden (2015-2019) and post-Rock Garden (2020-2022) marks. Pre- and post-Rock Garden marks are not significantly different for any of components of the Geological Mapping A course.

| 2015-19 dataset | Field | Field | Field | Report | Total |
|---|---|---|---|---|---|
| 2020-22 dataset | Field | Rock Garden | Field + Rock Garden | Report | Total |
| $t$-statistic | -0.45 | -1.57 | -1.44 | 1.55 | 0.89 |
| Degrees of freedom | 18.72 | 47.57 | 82.89 | 74.23 | 61.75 |
| $p$-value | 0.66 | 0.12 | 0.12 | 0.12 | 0.38 |

**Figures**

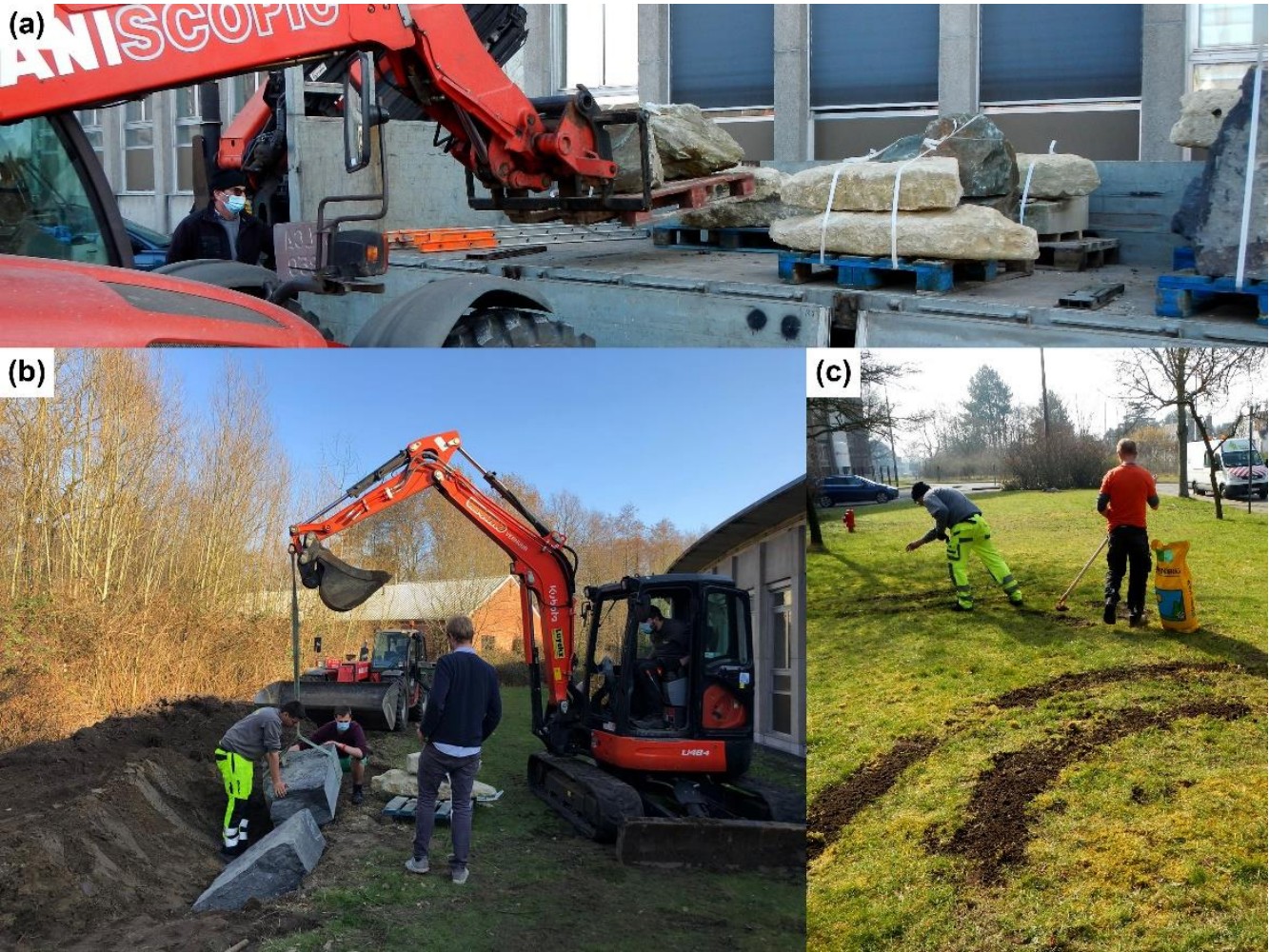

**Figure 1** Installation of the Rock Garden at Ghent University. **(a)** Rocks were delivered in coordination with Monument NV and unloaded with the help of the Ghent University DGFB. **(b)** Rocks were installed, under careful supervision, using both human and machine power. **(c)** After installation, green areas were reseeded with grass or wildflower seeds to ensure full recovery of the green spaces.

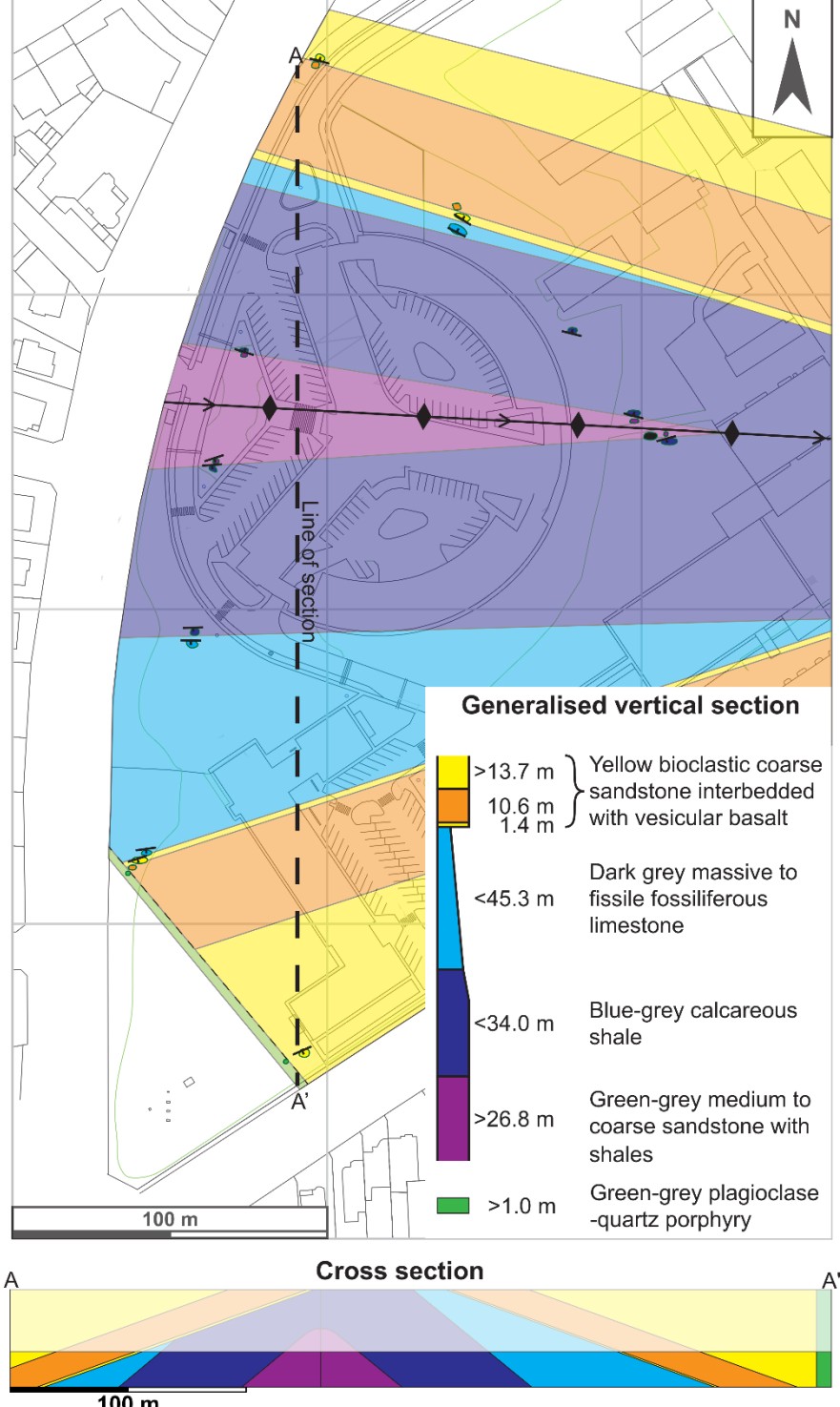

**Generalised vertical section**

| | | |
|---|---|---|
| >13.7 m | Yellow bioclastic coarse sandstone interbedded with vesicular basalt | |
| 10.6 m | | |
| 1.4 m | | |
| <45.3 m | Dark grey massive to fissile fossiliferous limestone | |
| <34.0 m | Blue-grey calcareous shale | |
| >26.8 m | Green-grey medium to coarse sandstone with shales | |
| >1.0 m | Green-grey plagioclase -quartz porphyry | |

**Cross section**

**Figure 2 A geological map of the Rock Garden.** The map is similar to that expected of students working on the Rock Garden, and their work includes producing annotated cross sections and generalised vertical sections alongside the main map.

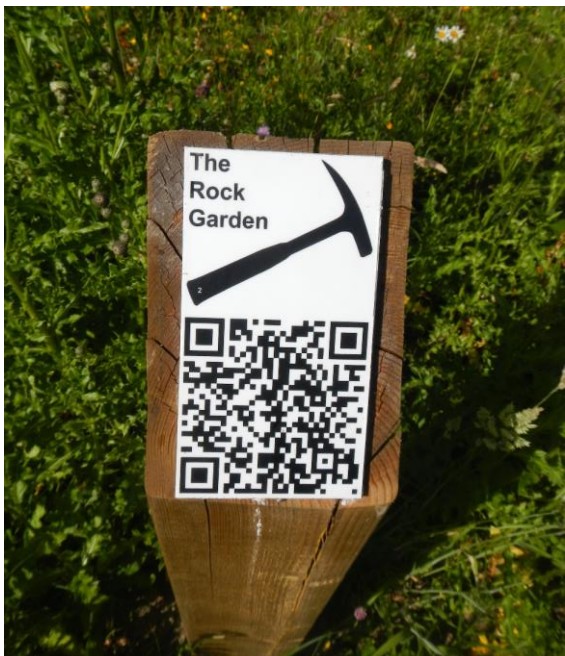

**Figure 3 QR codes are used on signs around the Rock Garden.** These QR codes link to short videos of field tests such as applying dilute hydrochloric acid to test for carbonate content. This means that students can think about when and why to perform these field tests whilst 570 ensuring the Rock Garden remains standing for future generations of students.

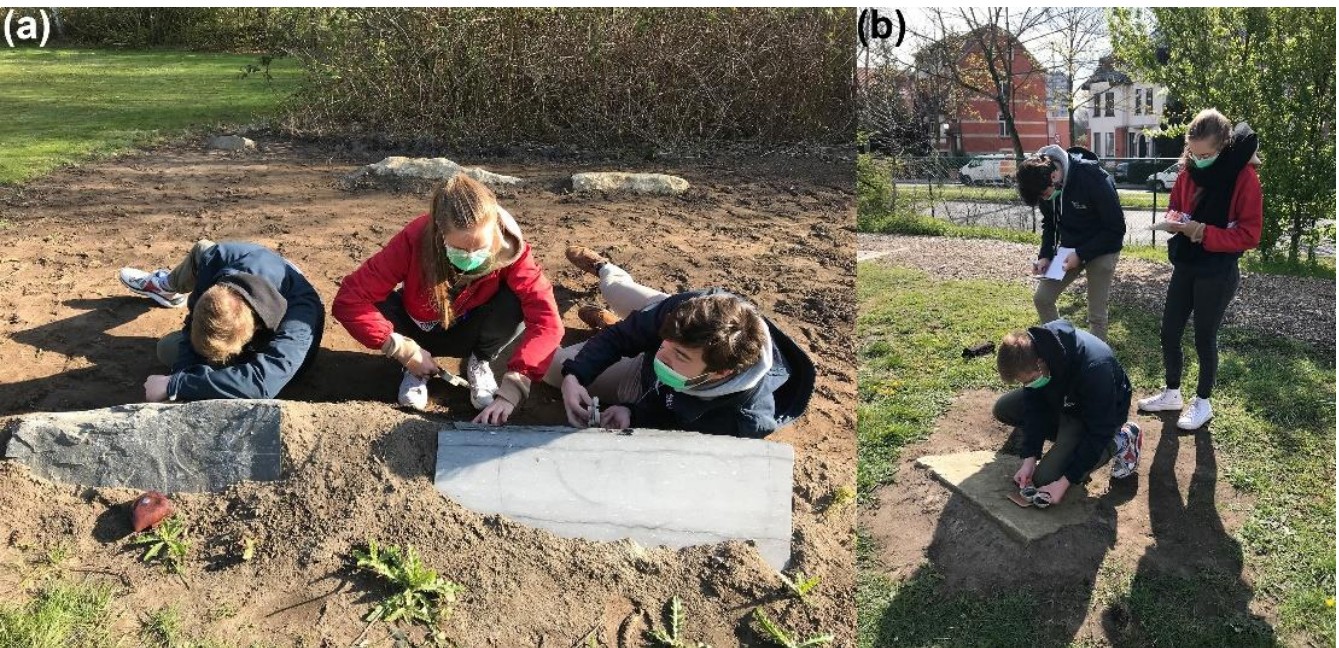

**Figure 4** Students practising using a compass clinometer on Rock Garden outcrops during their first mid-pandemic field course.

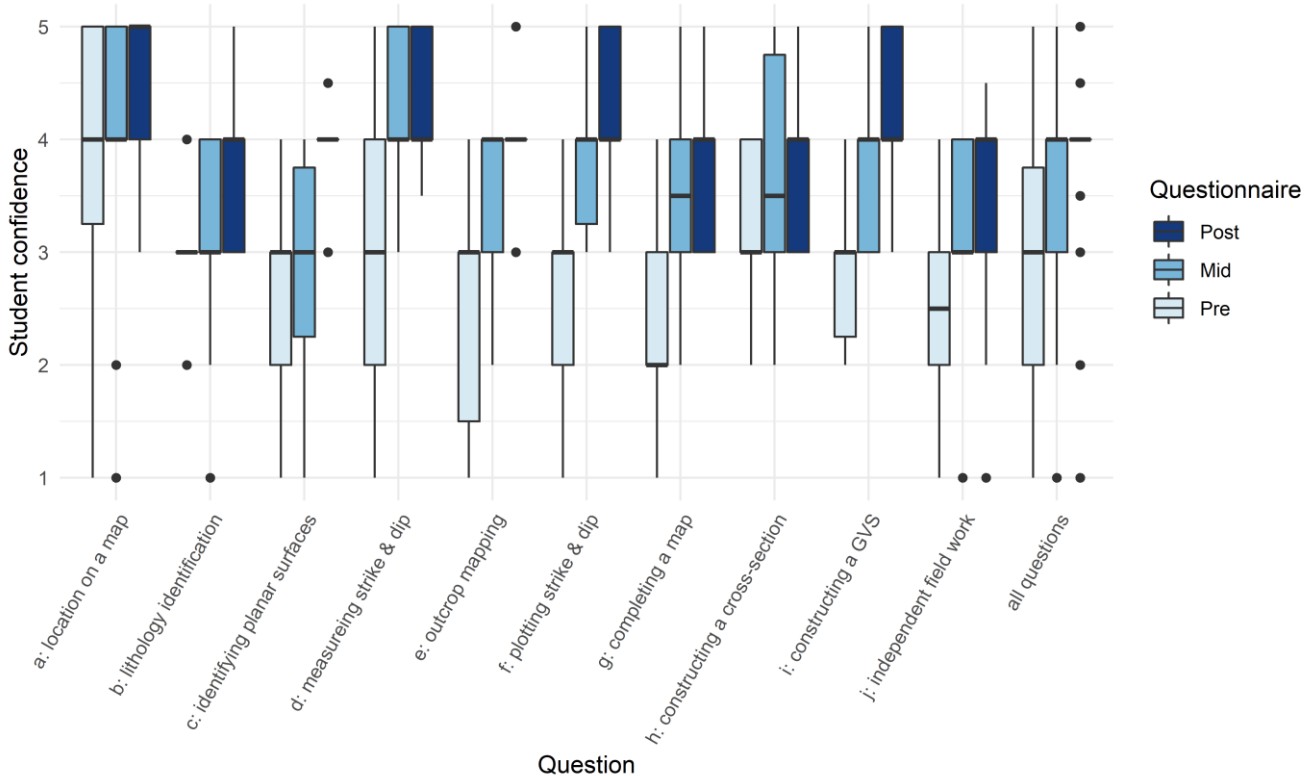


**Figure 5** Results of the student confidence self-assessment questionnaires. "Pre" = first questionnaire, before the Rock Garden exercise; "Mid" = second questionnaire, after the Rock Garden exercise; "Post" = third questionnaire, after the real world field course. In the first ("Pre") questionnaire, one student did not answer question *(d)* and one student did not answer question *(g)*; one student did not return the final ("Post") questionnaire; otherwise, all questions were answered. See Table 2 for the questions and Table 3 for a quantitative summary

of the data in this plot. GVS = generalised vertical section; "all questions" includes all question responses.

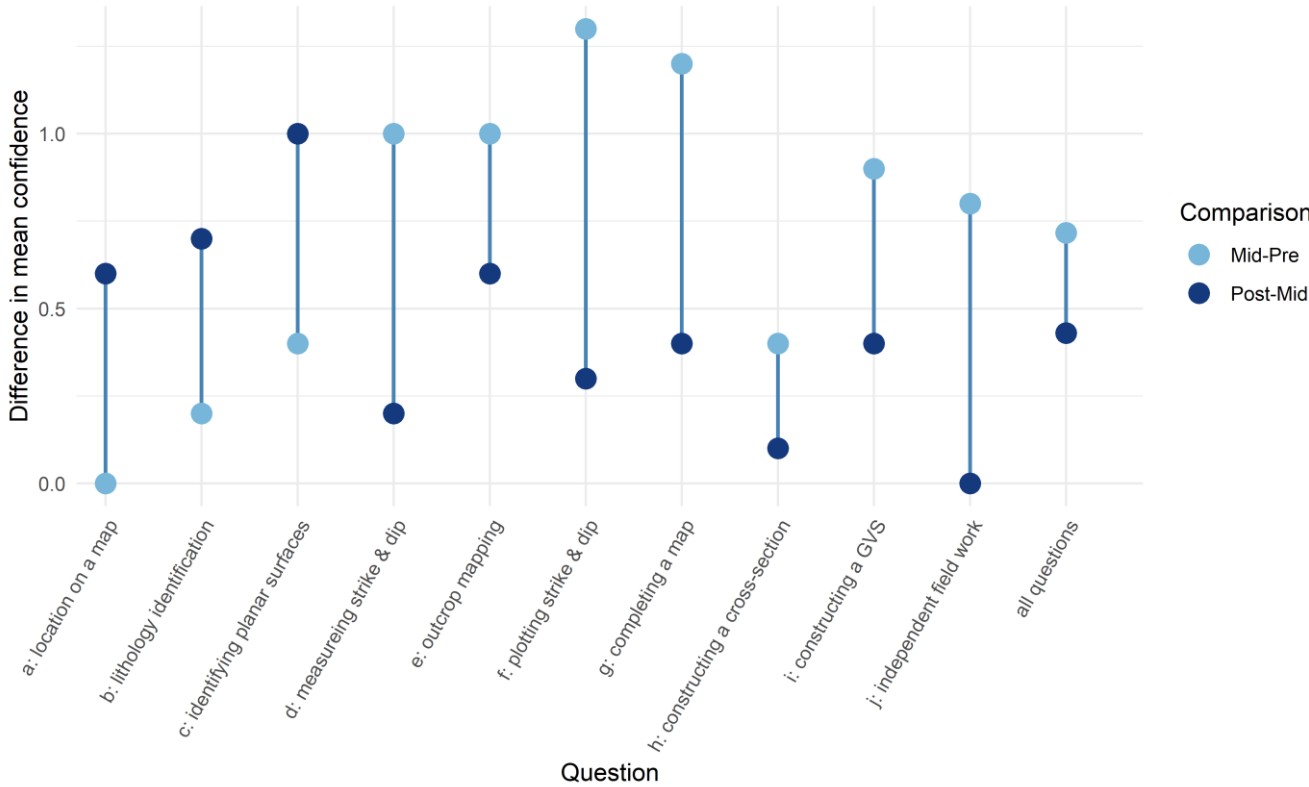

**Figure 6** Difference in student confidence between the mid- and pre-course questionnaires, and the post- and mid-course questionnaires. For questions *(a)*, *(b)*, and *(c)*, the greatest increase in confidence came from the real-world field course. The greatest increase in confidence for questions *(d)* to *(j)* came from the Rock Garden field course.

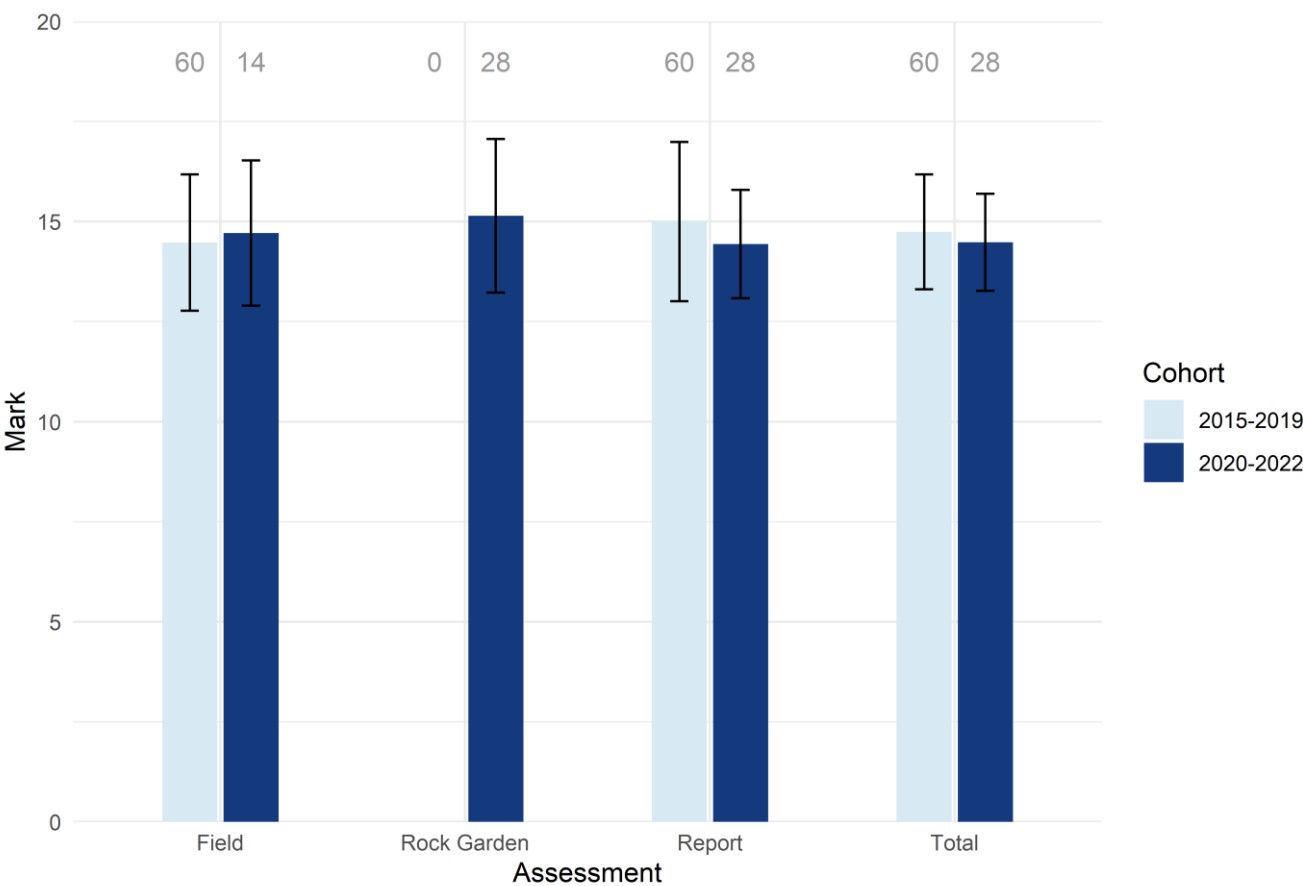

**Figure 7** Comparison of mean marks from the "Geological Mapping A" field course for 2015-19 (before the Rock Garden) and 2020-22 (the first two years using the Rock Garden). Numbers above each column are the number of individual marks included in each bar. Error bars represent one standard deviation.