# Peer review of "The Rock Garden: a preliminary assessment of how campus-based field skills training impacts student confidence in real-world field work"

_EGUsphere, 2023_

## Author Response (AR1)

School of Geography, Geology and the Environment

University of Leicester

University Road

LE1 7RH

UK

twonghearing@gmail.com

28.11.2023

Dear Dr Hillier

We are pleased to submit a revised version of our manuscript #EGUSPHERE-2023-766, now titled: "The Rock Garden: a preliminary assessment of how campus-based field skills training impacts student confidence in real-world field work".

We are grateful to you and the reviewers for your time and thoughtful comments on our manuscript. We have implemented each of the major revisions suggested by the reviewers, and we feel that these suggestions have substantially improved the manuscript overall. A full point-by-point response to the reviewers' comments can be found below.

In particular, we would like to highlight the two major changes to our manuscript. Firstly, largely in response to Reviewer 3, we have tried to better couch our research in the broader context of educational theory, especially that relating to Bloom's affective domain, and previous work on the role of the affective domain in geoscience education. Secondly, we have tried to better address the issue of the small sample size of our study raised by reviewers 1 and 3 both in the revised manuscript and in the revised title – emphasising the interesting future directions for research that our early analyses point to.

We thank you again for your consideration of our manuscript and look forward to hearing from you in due course.

Yours sincerely

Thomas W. Wong Hearing, on behalf of all authors

**Response to reviewers' comments**

**RC1**

**Overall**

This is a well written piece exploring an alternative to fieldclasses through the development of a campus 'Rock Garden'. An inability to travel to fieldclass locations is an issue that affected many education establishments globally during the pandemic, prompting a wider thinking on fieldwork in both undergraduate and postgraduate degree programmes within the Geosciences, but also within Environmental Sciences more broadly. It also highlighted the opportunity to revise how we think about fieldwork and potential improvements we can enact to enable greater inclusivity and equality of access to such opportunities.  The concept and idea presented here is interesting and valuable for others thinking about similar local resource developments.

**Response: We would like to thank Heather Sangster for her thoughtful and detailed review.**

The main challenge with this manuscript is the small sample size (n=14/13), which inevitably introduces high uncertainty in relation to the confidence that can be placed on the derived statistics. I think the authors would benefit from a larger sample size, and it might have been worth waiting for another year or two so that this could have been achieved.

It is also important to think about student cohorts and changes in outcome relative to other modules students have taken, to demonstrate whether changes might be a function of improved skills/learning, rather than different student cohort characteristics, particularly when the sample size is small.

**Response: We recognise that this is a small cohort study and that this brings statistical problems, and we try to carefully handle the statistical interpretations in view of this. There have been two practical constraints on this project, however. Firstly, we are limited by small student cohorts in geology at UGent. Secondly, two of the authors, including the lead author, have now left the university as they were on fixed-term PhD/post-doc contracts. Whilst not ideal, we consider it preferable to disseminate the results of the project thus far, whilst acknowledging that further work in this area is needed.**

**We have tried to address this point directly in our manuscript by revising the title to better reflect the preliminary nature of the conclusions, and in highlighting some of the further directions in which we would like to see research in this field go.**

It is important to note that there is a difference in confidence and skills, you identify the students have increased confidence, but do not demonstrate any notable shift in skills/learning, based on the marks, this is worthy of further reflection, even if this was something you try and capture in the future, or flag so others can consider this carefully.

**Response: We agree that the gain in student confidence/self-efficacy without similarly notable increase in skills, a point also raised by other reviewers, is worthy of further investigation. We have added a paragraph on this to the Discsussion. In particular, we reference the similarity of our results to those of Lundmark et al. (2020) who found an increase in self-efficacy was not matched by an increase**

**in assessed work. We do not want to overdo our discussion or conclusion from this small cohort study, but we highlight this as one of the most interesting areas of further research arising from our study.**

It would have been interesting to see if the same results (post-) would have been achieved without the Rock Garden exercise – is there a difference in endpoint confidence by including the fieldclass?

**Response: We agree that this is a fascinating question to address. Unfortunately, we do not have a suite of data from before the Rock Garden with which to test this, partly due to building the Rock Garden during the height of the COVID-19 pandemic to mitigate its impacts on student learning. Ideally, we would have collected such data in the years leading up to the Rock Garden initiative, similar to the approach of Waldron et al. (2016), and we would encourage others who are considering running a project like this to try to collect such baseline data before constructing their projects.**

A very interesting paper.

**Minor points**

Line 28, I would personally reorder 37 and 60 respectively, I think it might read more easily.

**Response: Done, and we have rephrased this sentence in light of the comment of RC3.**

L40-45, you can also include the inequalities caused for those with caring responsibility where residential not possible, and therefore these individuals are prohibited/discouraged from undertaking degrees/qualifications in the Environmental Sciences more broadly.

**Response: Done, and thank you for highlighting this.**

L126 Insert to: This allows students 'to' work

**Response: Done.**

L232 The sentence lacks clarity and needs to be revised, whilst the Rock Garden exercise saw the most rapid learning response, the real world experience still had a higher score, this needs to be clearly articulated.

**Response: We have rephrased to read: "In all other skills, there was a greater increase in confidence (a stronger self-efficacy response) after the Rock Garden exercise than after the real-world exercise. The highest confidence in all areas was achieved after the full course, including the real-world exercise (Figure 6)."**

L225-240 I would expect students to gain confidence after undertaking an exercise and with more experience, therefore some of these findings are not a surprise.

**Response: We agree and have added a sentence to emphasise this: "This is not particularly surprising as we would hope and expect students gain confidence from practising their skills; however, it is instructive to interrogate the differences in response between specific skills."**

**RC2**

My sincere apologies for the very late arrival of this review.

This is a good manuscript that describes an outdoor "garden" for use in geoscience training and education, and in outreach. The installation follows a developing tradition of outdoor geoscience teaching facilities located within universities that help to offset the many challenges involved in getting students into the field under more realistic circumstances. The authors highlight what we have found to be the principal advantage of our installation: it helps to bridge the gap for students between theoretical, classroom-based learning, and the valuable, but challenging, experience of distant real-world field locations where barriers of safety, unfamiliarity, accessibility, and cost must be overcome. The challenges of Covid-19 have added new urgency to these problems.

I have no problem with the scientific results presented, so my comments really come down to some questions that came up, comparing this study with our own (Waldron et al 2016), and a number of points where the wording could be improved for clarity.

**Response: We thank John Waldron for his thoughtful review.**

**Overall questions, comments.**

I'm interested in the results of your attempt to use actual student grades. In our study (Waldron et al. 2016) we didn't attempt this, for two reasons. First, our ethical guidelines made it very difficult to access student grades without informed consent; our perception was that our students were sufficiently nervous about their own grades that we would have many more students opting out if we asked for permission to access their grades. Secondly, and perhaps more seriously, having taught in the field we are aware that instructors instinctively adjust their grading (in areas that require subjective judgement calls in field situations) so as to take into account difficulties of weather, access, etc., which students encounter, and which vary from year to year. We were so skeptical that anything would show up in the marks that, given the challenges of gaining student consent, we didn't even bother to try this. I'm impressed that you tried, but not surprised that nothing significant showed up. (Note - this doesn't require anything to be changed in the paper - I'm just adding the comment for interest.)

**Response: We nearly didn't include this aspect within our study for the reasons outlined by the reviewer. However, it was considered to be in-line with the university's general ethical protocol and we considered it interesting to evaluate. We also agree that instructors instinctively adjust their grading dependent on various conditions where such marking is necessarily subjective, and we think this may be one reason for seeing a limited cognitive response to the Rock Garden exercise (see also our response to RC1). We have added further discussion on this point.**

Second, I'm curious about the geology implied by the cross-section shown. This would seem to imply that the older succession was folded, that the fold was truncated by a planar unconformity, and that the fold was then reactivated after the Eocene, folding the unconformity. A folded unconformity is quite an advanced concept for students who are just at the level of learning to measure strikes and dips. Was this a planned complexity, and did you consider setting the students the simpler problem of a fold truncated by a planar unconformity?

**Response: The geology is slightly more complex than we initially planned – settling of the rocks after positioning them slightly shifted some of the dips slightly and produced outcrops from which two episodes of folding could be inferred. We agree that this is perhaps too big an ask of students in an introductory exercise and are considering adjustments. We'd certainly recommend a simpler structure to others who look to build similar rock gardens.**

**Technical points for the authors**

Line number: given first for each comment.

25: American could refer to all of N and S America. Specify USA if that's what you mean.

**Response: We have rephrased this sentence to make clear that we mean "USA".**

50: The sentence beginning "We identified.." seems to be referring to some institution: maybe Ghent? This should be specified here; don't make the reader wait until line 57.

**Response: We have specified "At Ghent University…".**

51: Multi-word noun phrases like "more skills provision" can be hard to read. I recommend "more provision of field skills": much clearer with very little extra length.

**Response: We have rephrased this sentence.**

52: "brought in in response" - maybe "introduced" instead of "brought in". Also, "the global pandemic" is rather unspecific. Yes, I know what's meant now, but the paper still needs to be readable in 15 years time, so either specify the years or specify COVID-19.

**Response: We have changed it to "introduced" and specified "COVID-19".**

52 to 56: This is a long and convoluted sentence, especially "a valuable and viable alternative to bring..." where the reader expects "alternative to" to be followed by an actual alternative. Also "and which we also developed and used simpler versions of..." is quite clumsy. I'd suggest breaking the sentence after the first long list of citations. Then continue "These are a valuable and viable alternative to traditional field courses, bringing specific field areas to students at home or in the classroom (Bond etc..)". Then in a third sentence explain what you (Ghent) did along these lines, noting that it was simpler than some of the previous versions, and if possible citing some information about your virtual field trips...  (I'm assuming here that the initiative described by "which we also developed..." is a separate one from the Rock Garden, and was perhaps not particularly successful, but the description is frustratingly minimal. Either flesh it out with more details or just delete from "and which we..." onwards.

**Response: We have rephrased this section as suggested.**

107: outcrop sites (singular)

**Response: Done.**

128: Needs a comma after public. As written it implies that only some of the public have free access and others do not.

**Response: Done.**

135: Consider whether the phrase "Due to the limited size and number of outcrops.." really explains why there is no single correct answer. In my experience almost every real field area has more than one viable solution, unless perhaps there is 100% exposure, so the lack of a single solution is definitely not a difference from "real-world" mapping!

**Response: We agree with the reviewer's point and have rephrased the sentence to make it clear that we consider this to be a strength of the Rock Garden.**

160: The question asked here tests a hypothesis very close to that of Waldron et al (2016); I realize we are generously cited elsewhere, but I feel that this question was an innovation of ours; to deal with the difficulty of assessing changes in actual student performance, we assessed student perception or their own performance (metacognition).

**Response: Also in response to reviewer 3, we have rephrased the Aims & Hypotheses section to better couch our work and hypotheses in the context of previous studies including Waldron et al. (2016). We would note that we follow a similar approach to Waldron et al. (2016) but ask a slightly different question of the students, which is how confident they feel in applying their skills. This is related, but not identical, to how well students think that a course has prepared them to conduct work on a future course, which is the question asked by Waldron et al. (2016). Because of this, we considered it more suitable to have an expanded explanation, further developed in the Discussion, focused on the work of Waldron et al. (2016) and Benison (2005), but not to include it in the short description of each hypothesis.**

194: The hyphenated word post-Rock breaks the flow because the sense is post-(RockGarden). Options are to write "post-Rock-Garden" or "post-installation", or use the more wordy (but more understandable) "data collected after the Rock Garden exercise".

**Response: We used this phrasing to maintain consistency with the tables and figures. We understand the reviewer's comment but would prefer to keep this as-is, because the use of "post-" here and elsewhere alongside the "pre-" and "mid" prefixes succinctly describe the questionnaire rounds.**

199: Hyphenate small-cohort because it's used as an adjective.

**Response: Done.**

222: and elsewhere: "Reject the null hypothesis.." The null hypothesis is never actually stated in those cases where it is rejected. The text only mentions the alternative hypothesis that was upheld by the rejection. I recommend explicitly adding the null hypotheses somewhere — perhaps in the list , lines 203 to 205, where the null hypotheses could be put in parentheses following each hypothesis to be tested.

**Response: Thank you for pointing this out. We have added explicit statements of the null hypothesis to each of the hypotheses sections.**

230: These nested parenthesis are code-like and difficult to read. Consider using one-word shorthands for the different survey questions instead of [b] and [c].

**Response: Done, here and elsewhere.**

267: "less" should be "fewer" unless you are faithfully translating from similarly incorrect Dutch

**Response: Done.**

305: (and elsewhere). In British English this would be practising; the noun has a c and the verb an s.  The form used here follows the US convention of using -ice for both the noun and the verb. I mention it here because you may wish to check the journal style guidelines and be consistent.

**Response: Done.**

**RC3**

**General comments / scientific issues:**

This is a very well written and presented paper that evaluates the impact of an on-campus rock garden on students' confidence in field skills. This is a novel approach and I really like that the study is framed in the context of addressing barriers to access and inclusion, and also in response to travel restrictions imposed during the pandemic. The following comments are for consideration alongside those made by the other reviewers, which I broadly support and see no need to repeat.

**Response: We thank Alison Stokes for this thoughtful and considered review. Regarding points raised by other reviewers that Stokes supports, we hope that she finds our responses to these to be satisfactory.**

The authors have clearly gone to some effect to generate a dataset but I query whether this is 'research'. Acknowledging that research and evaluation exist along a continuum, I would expect a research article to be more rigorously framed with respect to the wider literature (beyond geoscience education) and a guiding theoretical / conceptual framework, and to present more robust findings and conclusions. This is a preliminary evaluation of a curriculum innovation which has been carefully designed and is well reported, and which provides a fantastic springboard for a more in-depth research study – but the sample size is problematic and I echo the other reviewers' comments about confidence in the findings (or lack of). I would recommend that the title of the paper is tweaked accordingly as there is clearly some uncertainty around the apparent gains in confidence.

**Response: We agree that the small sample size is a clear limitation of this study, but note that we are constrained by small cohorts in geology at UGent. We have amended the title of the manuscript to better reflect this. We have also reframed some of the text and added further statements throughout the manuscript to caution against over-interpretation of the statistical analyses we present, and to highlight areas where our results point towards avenues for future research.**

I'm interested to know the bases for the hypotheses. What are the assumptions that are being made here, and how are these informed by the literature? Some consideration / discussion of the wider literature around confidence gains / self-efficacy would really help to strengthen this. There is evidence to show that students' confidence in being able to accomplish a task is influenced by how they feel, and that during fieldwork students' feelings / emotions can be influenced by a whole range of factors such as weather, time of day, the end-goal (are they being assessed etc.) – worth looking into literature around the affective domain, and also the link between confidence and learning gains.

I'd also recommend placing the findings in a broader context by bringing some of the literature into the discussion. Some interesting examples of other, similar initiatives are introduced in section 2.1 - has there been any similar evaluation of impacts on student learning? If so, what did they find out and how do your findings contribute to the evolving body of knowledge about the potential for rock gardens to support / enhance student learning?

**Response: We think these are excellent suggestions and have implemented both. We have expanded both the Aims & Hypotheses and Discussion sections to better site our work in the broader literature**

of the affective domain and its link to the cognitive domain, with a focus on studies pertaining to geoscience education. In particular, we have tried to highlight how our work fits within the body of work relating geoscience field work to the affective domain.

Overall, our study contributes to the broader corpus of literature that shows a positive affective response from field work activities. In our work this is both as a semi-quantitative (albeit with small numbers) positive self-efficacy response, and as an overall positive impression left by field work on the students. We have substantially expanded both the Aims and Discussion sections on the topic of 'Rock Garden' studies, notably including more on the work of Waldron et al. (2016) and Benison (2005). Our work qualitatively supports the findings of both Benison (2005) and Waldron et al. (2016) that students find a campus-based field resource helpful training in general, and that it is beneficial for subsequent field work (Waldron et al. 2016). Our work also provides some additional nuance about when self-efficacy gains may be made in the learning process (i.e. the strongest self-efficacy gain may occur during the on-campus training).

Perhaps more interesting, however, is the similarity of our results with those of Lundmark et al. (2020) who also found no improvement in assessment grades despite an increase in self-efficacy following the implementation of a novel education aid in geoscience field work. Whilst trying not to overstate our conclusions, we suggest that this is an area that would be particularly interesting to study further.

All this said, we are explicitly wary of taking our discussion too far, given the sample size, and try to emphasise that these are areas that would benefit from substantial further study.

Overall, this is a nice study that provides some preliminary evaluation of on-campus rock gardens as a learning tool – on this basis it will be a good contribution to the literature. However, there needs to be more rigorous follow-up investigation to support any substantial claims about impact.

Response: We are grateful for this thoughtful review and agree that our study should be seen as providing a springboard to further research on the impacts of artificial field courses on students' skills and self-efficacy acquisition.

**Technical corrections:**

L23-24: note that the Geological Society of London has updated its accreditation requirements and no longer specifies a number of field days that students must complete (see https://www.geolsoc.org.uk/~/media/shared/documents/education%20and%20careers/University%20Accreditation/2023%20documents/Introduction%20and%20Guide%202023.pdf?la=en).

Response: We have rephrased this.

**CC1**

Comment:

Dear authors,

I think the concept presented here is very interesting and potentially very valuable in teaching (and removing barriers for those who may not even be able to attend conventional field courses).

**Response: We are grateful to Sebastian Mutz for taking the time to provide helpful comments on our manuscript.**

I do have some concerns about the sample size (and some of the larger p-values), however. The authors do acknowledge that problem and I think the authors mostly treat this problem very well and with commendable transparency. That said, as someone who is interested in implementing such an approach, I would want to see more thorough testing of it (as part of future studies at least). I do not think this is absolutely necessary for a pilot study, but I do think the manuscript would benefit from (a) adopting a more careful tone in the discussion of results (/toning down the claims a little), and (b) placing a little more emphasis on encouraging the testing of this type of approach.

**Response: We are delighted to hear of more people interesting in adopting this approach and warmly encourage Sebastian in doing so! We are aware of the limitations that a small sample size places on our study – a problem noted by other reviewers – and this is somewhat inevitable due to the small cohort sizes of geology at UGent. We try to treat our results cautiously, and have revised our text in several places, including the title, to better reflect this.**

I would also recommend the inclusion of skill-based tests of progress, because confidence does not necessarily reflect skill.

**Response: We agree, certainly, that evaluating how a resource like the Rock Garden influences the progress of skills acquisition is an interesting and worthwhile study. However, our primary aim here was to assess how students' confidence in applying skills that we knew they already. From the outset, we knew (from their work) that the students do have these practical skills, acquired through previous courses (though skills can always be further developed and honed). Our aim here was to understand whether using the Rock Garden as a teaching aid would help students to become independent field scientists by increasing their confidence in working in the field (their self-efficacy). We think it would be challenging to get ethical approval and student support for including more granular marks data, not to mention mitigating the necessarily subjective nature of marking practical field work. We also feel sure that drawing meaningful conclusions on the impact of the Rock Garden on specific field skills attainment would require larger cohorts than we typically have. With that in mind, we would be happy to see this develop in future work by others, but are not sure that this is something we can do at Ghent University.**